# Drop, Swap, and Generate: A Self-Supervised Approach for Generating Neural Activity

**Ran Liu**[*]
Georgia Tech

**Mehdi Azabou**
Georgia Tech

**Max Dabagia**
Georgia Tech

**Chi-Heng Lin**
Georgia Tech

**Mohammad Gheshlaghi Azar**
DeepMind

**Keith B. Hengen**
Washington Univ. in St. Louis

**Michal Valko**
DeepMind

**Eva L. Dyer**[*]
Georgia Tech

## Abstract

Meaningful and simplified representations of neural activity can yield insights into *how* and *what* information is being processed within a neural circuit. However, without labels, finding representations that reveal the link between the brain and behavior can be challenging. Here, we introduce a novel unsupervised approach for learning disentangled representations of neural activity called *Swap-VAE*. Our approach combines a generative modeling framework with an *instance-specific alignment* loss that tries to maximize the representational similarity between transformed views of the input (brain state). These transformed (or augmented) views are created by dropping out neurons and jittering samples in time, which intuitively should lead the network to a representation that maintains both temporal consistency and invariance to the specific neurons used to represent the neural state. Through evaluations on both synthetic data and neural recordings from hundreds of neurons in different primate brains, we show that it is possible to build representations that disentangle neural datasets along relevant latent dimensions linked to behavior.

## 1 Introduction

In the brain, the coordinated actions of groups of neurons are responsible for encoding sensory inputs and movements, as well as all processing and manipulation in between (1; 2; 3; 4; 5). Understanding what different populations of neurons are doing and how they work together to encode their inputs is a primary goal of neuroscience (6).

When successful, representations learned from populations of neurons can provide insights into how neural circuits work to encode their inputs and produce behavior, and allow for robust and stable decoding of these correlates. Over the last decade, a number of unsupervised learning approaches have been introduced to build representations of neural population activity without knowledge of specific labels or downstream decoding objectives (7; 8; 9; 10; 11; 12; 13). Such methods have provided exciting new insights into the stability of neural responses (14), individual differences (10), and remapping of neural responses through learning (15). However, without labels or additional inputs to guide the network, learning representations that allow *different sources of variability to be distinguished* is still a major challenge (16; 17).

---

[*]Contact: rliu361@gatech.edu, evadyer@gatech.edu. Project page: https://nerdslab.github.io/SwapVAE/.

35th Conference on Neural Information Processing Systems (NeurIPS 2021).

Here, we develop a novel unsupervised approach for disentangling neural activity called *Swap-VAE*. Our approach is loosely inspired by methods used in computer vision that aim to decompose images into their *content* and *style* (18; 19; 20; 21; 22): the representation of the content should give us the abstract "gist" of the image (what it is), and the style components are needed to create a realistic image (or, equivalently, they capture the variation in images with the same content). To map this idea onto the decomposition of brain states, we consider the execution of movements and their representation within the the the brain (Figure 1, Right). The content in this case may be *knowing where to go* (target location) and the style would be the *exact execution of the movement* (the movement dynamic). We ask whether the neural representation of movement can be disentangled in a similar manner.

To identify the *content* within our neural recording, we use a self-supervised approach: we apply a variety of transformations to the input data (observed firing rates) which we hypothesize to be content-preserving, and train the representation to be invariant to these manipulations. These transformed (or augmented) views are created by dropping out neurons and jittering samples in time, which intuitively should lead the network to a representation that maintains both temporal consistency and invariance to the specific neurons used to represent the neural state. In addition to this instance-specific alignment loss, we also encourage the network to reconstruct the original inputs using a regularized variational autoencoder (beta-VAE) (23; 24) that has access to both the content variables and another set of variables in the model that encodes the *style*. We show that through combining our proposed self-supervised alignment loss with a generative model, we can learn representations that disentangle latent factors underlying neural population activity.

We apply our method to synthetic data and publicly available non-human primate (NHP) reaching datasets from two different individuals (25). To quantify how effectively our method disentangles these datasets, we propose several general-purpose measures of representation quality, which characterize the extent to which variation in content and style is isolated by the two different spaces. We show that by using our approach, we can effectively disentangle the behavior and dynamics of movement without any labels. Our model thus strikes a nice balance between view-invariant representation and generation.

Our specific contributions are as follows:

- In Section 3, we propose a generative method, *Swap-VAE*, that can both (i) learn a representation of neural activities that reveal meaningful and interpretable latent factors and (ii) generate realistic neural activities.

- To further encourage disentanglement, we introduce a novel latent space augmentation called *BlockSwap* (Section 3.3), where we swap the content variables between two views and ask the network to predict the original view from the content of a different view.

- In Section 3.4, we introduce metrics to quantify the disentanglement of our representations of behavior and apply them to neural datasets from different non-human primates (Section 4) to gain insights into the link between neural activity and behavior.

## 2 Background and Related Work

### 2.1 Variational autoencoders and their application in neural data analysis

Variational auto-encoders (VAEs) (26) are a popular deep generative learning framework used to generate and denoise data. Let $\mathbf{x}$ and $\mathbf{z}$ denote the data and the latent variables, respectively, where $\mathbf{z} = q_\phi(\mathbf{x})$ is the latent representation extracted from $\mathbf{x}$ by the encoder $q_\phi$. The usual objective of probabilistic generative models is to maximize the log evidence of the observed data $\max_\theta \log p_\theta(\mathbf{x}) = \int p_\theta(\mathbf{x}|\mathbf{z})p(\mathbf{z})$ based on the parameterized model $p_\theta$ (called the decoder). VAEs and their variants instead optimize a tractable lower bound on the original objective, which is also well-known as the evidence lower bound (ELBO) in variational Bayesian inference (27),

$$\log p_\theta(\mathbf{x}) \geq \mathbb{E}_{z \sim q_\phi(\mathbf{z}|\mathbf{x})}[p_\theta(\mathbf{x}|\mathbf{z})] - \beta D_{KL}(q_\phi(\mathbf{z}|\mathbf{x})||p(\mathbf{z})) =: \mathcal{L}_{\theta,\phi}^{\text{VAE}}, \tag{1}$$

where the encoder $q_\phi$ is trained to approximate the Bayes posterior $p_\theta(\mathbf{z}|\mathbf{x})$, $p(\mathbf{z})$ is the prior over the latent variables, $D_{KL}$ is the Kullback–Leibler divergence, and $\beta \geq 1$ is a trade-off parameter. Specifically, the standard VAE is obtained when setting to $\beta = 1$, while $\beta > 1$ corresponds to the beta-VAE (23). A larger value of $\beta$ imposes stronger implicit regularization on the posterior of

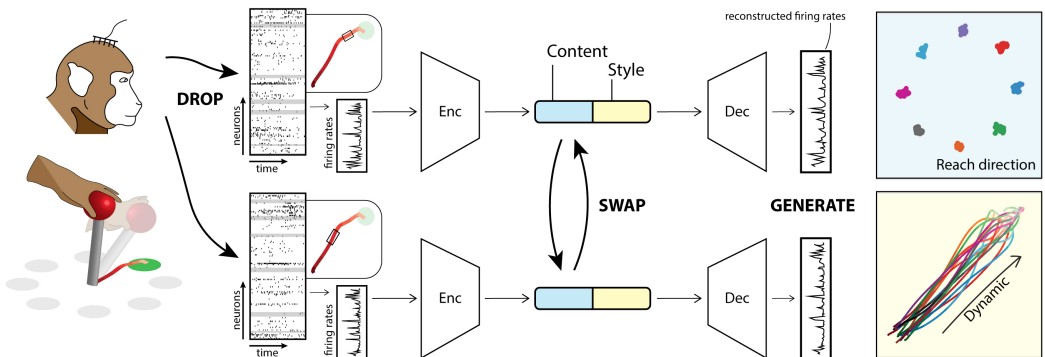

Figure 1: *Overview of approach.* On the left, we show an illustrative figure of the motor cortex reaching datasets, where rhesus macaques are trained to reach one of eight targets while their brain state (activity of many neurons at an instance in time) is recorded. Different views of a brain state are generated by dropping out neurons and exchanging samples that are close in time. The encoder (Enc) of the model extracts and partitions the latent variables of the two views into a content space and a style space. An instance-specific loss is applied to the content representations of two augmented brain states to encourage alignment, while neural activity is reconstructed through a decoder (Dec) using both parts of the latent space. To further enhance alignment in the content space, we introduce *BlockSwap*: the content variables of the representations of the two augmented views are swapped before being passed through the decoder. To the right, we show how the behavior of the rhesus macaques movements can be disentangled, where a reach to a target can be decomposed into the direction of the movement (blue) and its underlying dynamics (yellow).

latent variables to align with the prior, which empirically induces more disentanglement and thus interpretability in the learned representations (24; 28; 29).

Recently, a number of generative modeling techniques based upon VAE have been applied to neural data. LFADS and more recent extensions of this model (10; 9), use a sequential VAE (30; 31) to estimate neural population dynamics and show that this model can faithfully reconstruct single trial firing rates. More recently, pi-VAE (7) was proposed to learn representations of neural activity using a simple MLP encoder to capture information in each neural state vector (firing rate of $d$ neurons) independently. They show that, even with a simplified architecture that treats each sample independently, it is possible to learn an identifiable model that can directly link behaviors to neural responses. In our work, we also use a MLP encoder similar to (7), as we do not explicitly incorporate temporal structure into our architecture. Instead, we train the model to disentangle the content (target) from the dynamics without regularization from a more complex architecture.

## 2.2 Instance-specific alignment and self-supervision

Recent self-supervised learning (SSL) methods have made impressive advances, now rivaling (or in some cases surpassing) supervised methods (32; 33; 34; 35). To build representations, these approaches aim to maximize the similarity across multiple augmented "views" of the same sample (positive examples) (36; 37; 34). Thus, instead of providing labels to regularize the latent representations, we instead supply augmented versions of the same example and align their latent representations. Intuitively, training the network to *align the instance-specific views* where the semantic content is preserved, will encourage it to learn meaningful representations, which can then be used to solve downstream tasks (38).

Building on these successes, (11) recently showed how instance-specific alignment (using only positive examples) can be applied to multi-neuron recordings. This work shows that by combining simple augmentations, including: dropout, temporal shift (selecting nearby points in time), and additive noise, with a dual network, it is possible to learn representations that are useful for decoding behavior. In this work, we further show that coupling this type of alignment approach with a generative model, we can build networks that achieve both good approximation quality and disentanglement, all without the need for a dual (online/target) network. This greatly simplifies our architecture and optimization approach.

# 3 Methods

In this section, we introduce our self-supervised approach for generative modeling of neural data. A PyTorch (39) implementation is provided here: `https://nerdslab.github.io/SwapVAE/`.

## 3.1 Model for neural datasets

Throughout, we consider collections of $d$ neurons that have been spike sorted and binned to compute an estimate of the firing rate of all neurons at $N$ distinct time points (bins). This results in an input vector $\mathbf{s}_i \in \mathbb{R}^d$ as an observation at each time point. Let $\mathcal{D} = \{\mathbf{s}_1, \dots, \mathbf{s}_N\}$ denote the neural state vectors generated by this binning process. Let $k$ denote the dimension of the latent space.

## 3.2 Unifying instance-specific alignment and generative modeling

As we describe in the introduction, our aim is to build a decomposable picture of brain states. To do this, we will leverage the principles of *self-supervision* to build a view-invariant representation and use this as the building block for our generative modeling approach.

Our goal is to learn two functions, an *encoder* $f : \mathbb{R}^d \to \mathbb{R}^k$ and *decoder* $g : \mathbb{R}^k \to \mathbb{R}^d$. Let $\mathbf{x}_1 = t_1(\mathbf{s})$ and $\mathbf{x}_2 = t_2(\mathbf{s})$ denote the views generated after applying two random transformations $t_1, t_2 \sim \mathcal{T}$ to a sample $\mathbf{s}$ from our dataset $\mathcal{D}$. Let $\mathbf{z}_1 = f(\mathbf{x}_1), \mathbf{z}_2 = f(\mathbf{x}_2)$ denote the representations of both views in the network. To decouple the factors of our latent representations, we divide the latent space into two parts, $\mathbf{z}_1 = [\mathbf{z}_1^{(c)}, \mathbf{z}_1^{(s)}]$ and $\mathbf{z}_2 = [\mathbf{z}_2^{(c)}, \mathbf{z}_2^{(s)}]$, with $\mathbf{z}_1^{(c)}$ and $\mathbf{z}_1^{(s)}$ modeling the behaviour styles and intrinsic neural contents, respectively. $\widehat{\mathbf{x}}_1 = g(\mathbf{z}_1)$ and $\widehat{\mathbf{x}}_2 = g(\mathbf{z}_2)$ are the reconstructions of both views obtained after passing them through the decoder.

To encourage alignment of the views through the encoder while also solving our generative modeling objective, we propose the following loss:

$$\min_{f,g} \ \sum_{i=1,2} \underbrace{\mathcal{L}_{\text{rec}}(\mathbf{x}_i, g(\mathbf{z}_i))}_{\text{Reconstruction loss}} + \beta \sum_{i=1,2} \underbrace{D_{KL}(\mathbf{z}_i^{(s)} \parallel \mathbf{z}_{i,\text{prior}}^{(s)})}_{\text{Regularization - style space}} + \alpha \ \underbrace{\mathcal{L}_{\text{align}}(\mathbf{z}_1^{(c)}, \mathbf{z}_2^{(c)})}_{\text{Alignment - content space}}, \tag{2}$$

where the alignment loss $\mathcal{L}_{\text{align}}$ encourages two views to be close (here we used a normalized L2-distance), $\alpha$ and $\beta$ are hyperparameters that determine the tradeoff between alignment and reconstruction, and the KL divergence terms measure the deviation between the style latent variables and the prior $\mathbf{z}_{i,\text{prior}}^{(s)}$ which we set to be the isotropic Gaussian $\mathcal{N}(\mathbf{0}, I)$. In our experiments on neural datasets, we choose the reconstruction loss $\mathcal{L}_{\text{rec}}$ to be the Poisson loss (10; 40). Further details on the method and our implementation is provided in Appendix A.

## 3.3 *BlockSwap*: A novel latent space augmentation for disentanglement

To further improve disentanglement in our model, we propose the following novel *latent space augmentation*, which basically exchanges the content information (block of variables) between two augmented views while keeping their style constant. We refer to this latent augmentation as *BlockSwap* as it holds one part of the representation consistent and then swaps a different subset of latent variables between two augmentations. Specifically, after generating the representations $\mathbf{z}_1 = [\mathbf{z}_1^{(c)}, \mathbf{z}_1^{(s)}]$ and $\mathbf{z}_2 = [\mathbf{z}_2^{(c)}, \mathbf{z}_2^{(s)}]$ for each view, we also generate their content-swapped versions $\widetilde{\mathbf{z}}_1 = [\mathbf{z}_2^{(c)}, \mathbf{z}_1^{(s)}]$ and $\widetilde{\mathbf{z}}_2 = [\mathbf{z}_1^{(c)}, \mathbf{z}_2^{(s)}]$. To encourage disentanglement, we propose to replace the previous reconstruction term by adding the loss over the swapped representations:

$$\mathcal{L}_{\text{rec}}^{\text{swap}} = \sum_{i=1,2} \underbrace{\mathcal{L}_{\text{rec}}(\mathbf{x}_i, g(\widetilde{\mathbf{z}}_i))}_{\text{Swapped content}} + \underbrace{\mathcal{L}_{\text{rec}}(\mathbf{x}_i, g(\mathbf{z}_i))}_{\text{Original}}. \tag{3}$$

When we use this loss, we can also consider removing the alignment term in Equation 2 and simply couple this reconstruction loss with the regularization KL term on the style space (see Section 4.3). In practice, the reparameterization trick is performed before the swapping of content and style variables.

### 3.4 Representational quality and disentanglement metrics

Determining whether a representation captures the underlying factors of interest is, in general, very challenging (16; 41; 42). In neuroscience this is certainly true. We will consider two main measures of representation quality and disentanglement to guide our investigation.

**Multi-task disentanglement score.** To confirm that our method effectively disentangles the represented behavior of neural signals from the dynamics, we need to measure the extent to which latent variables respond to one or the other with specificity. Concretely, let $\mathbf{z}$ denote a $k$-dimensional latent representation, and let $y_c, y_s$ be discrete values that label the reaching direction and dynamics that correspond to $\mathbf{z}$, respectively. For each latent variable $\mathbf{z}_i$ ($1 \leq i \leq k$), we assess the degree to which it is aligned with the content of the behavior ($y_c$) or the temporal structure ($y_s$). Computing the covariance score for a particular latent variable $\mathbf{z}_i$ consists of three steps:

1. Compute the variance when changing $y_c$ with fixed $y_s$, and average over values of $y_s$.
2. Compute the variance when changing $y_s$ with fixed $y_c$, and average over values of $y_c$.
3. Compute the absolute difference of the two variances. This is the score for $\mathbf{z}_i$.

Intuitively, if the score is large, then $\mathbf{z}_i$ changes more dramatically in response to one parameter than the other, so it displays specificity, while if it is low then the amount that $\mathbf{z}_i$ changes is nearly the same. Averaging across all latent variables after normalization gives a final score, which provides a measure of how disentangled the entire representation is.

**Linear readout from representation layer.** To further quantify the representation quality and stability of representations in downstream decoding, we use a linear readout strategy employed frequently in self-supervised learning approaches (32; 34). In particular, we will train the model on our training dataset, freeze the weights in the network, and then train a linear layer to decode the reach directions from the output of the encoder. However, because we are also interested in disentanglement, we will consider the prediction of two different class labels from either the full, content, or style factors in the network. When decoding either reach directions $y_c$ or temporal structure $y_s$, we will retrain the linear weights but keep the representation fixed.

As in (11), we quantify reach decoding with two scores, acc and delta-acc, for the linear decoding accuracy on reach direction. Consider the reaching task as a regression over a circle with a total of $l$ discrete labels, we count the decoded angle that falls within $[(2i-1)\pi/l, (2i+1)\pi/l]$ as the correct classification in acc, and that falls within $[(2i-1.5)\pi/l, (2i+1.5)\pi/l]$ as the correct classification in delta-acc. The two scores both provide a measure of the representation quality in terms of the precision of reach direction decoding. The time decoding accuracy is computed similarly, but in this case, the goal is to predict how far into each reach each sample is.

## 4 Experiments

The usefulness of this method depends on its ability to decompose neural data into meaningful latent factors. To quantify this, we devised experiments to reveal three desirable properties:

1. Performance on downstream classification tasks (linear separability of the representations).
2. Faithfulness of the latent space structure to the ground-truth structure of the task.
3. Separability of the content and style components across the respective latent subspaces.

### 4.1 Synthetic experiments

We first trained and evaluated our model on a synthetic dataset that was designed to resemble our neural datasets. The data are generated and the experiments are designed following the approach used in (43; 7).

**Synthetic reaching dataset.** We generated latent variables from a 2-dimensional mixture of four Gaussians, where the $i$th component Gaussian has mean $(5\sin u_i, 5\cos u_i)$ and variance $(0.6 - 0.3|\sin u_i|, 0.3|\sin u_i|)$ and $u_i$ is uniformly sampled from the interval $[\frac{i\times\pi}{2}, \frac{(i+1)\times\pi}{2}]$, $i \in \{0, 1, 2, 3\}$. For each sequence, we randomly sampled $l = 4$ data points within each cluster and order them in a clockwise manner to create consistent dynamics in the latent space (as shown in Figure 2 on the left). The formed sequences were fed to a RealNVP network (44) to generate 100-dimensional Poisson observations of the firing rates. The generated synthetic firing rates are then shuffled and

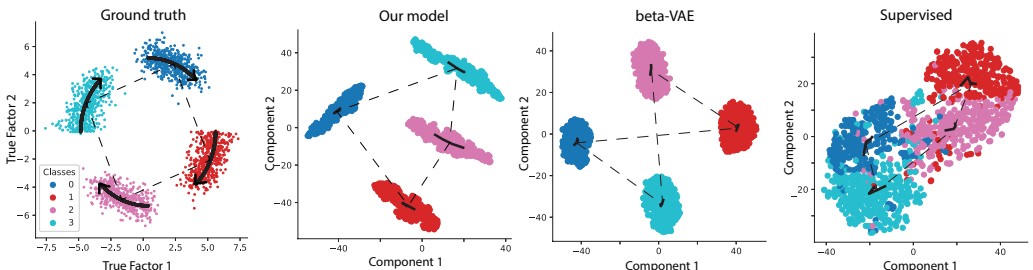

Figure 2: *Synthetic Experiments.* On the left, we show the ground truth latent space and their dynamics, from which the firing rate is generated in a clockwise manner. To the right, we show the results of our model, a beta-VAE, and a supervised model. Our model recovers both the discrete classes and the sequential structure present within this synthetic dataset.

split into 80% training set and 20% test set. The ground truth latent distributions and the generated results are shown in Figure 2. Ideally, we want the model to capture and disentangle both the discrete groups/clusters $y_c$ and the dynamics of the sequence (which are implicitly encoded via the RealNVP network).

**Results on synthetic datasets.** To compare the representational power of our approach, we applied our model, the beta-VAE, and a supervised decoder (trained to predict $y_c$) to these synthetic datasets. All models have the same backbone with a latent space of 32-dim, which in our model is split into a content space of 16-dim and a style space of 16-dim. Each model is trained on the training set for 100,000 iterations, and optimized using Adam with a learning rate of 0.0005 (further details on model selection and hyperparameter optimization in Appendix A). When we examined the representations formed by each of these models, we found that *Swap-VAE* was very effective at both preserving the sequence dynamics (as highlighted by the connection between class centroids) as well as separating the different target classes (Figure 2). From the figure, we can see that while the beta-VAE successfully separated different clusters, the ordering of clusters is not aligned with the true dynamics (the black line), while the elongated distribution formed by our model more accurately reflected the true distribution of each component, regardless of the noise. Using the metrics described in Section (3.4), we further confirmed that our model provides good disentanglement, producing a multi-task disentanglement score of 0.93. The corresponding score for the beta-VAE and supervised model were 0.46 and 0.12, respectively.

## 4.2 Experiments on neural datasets

After testing the model on synthetic datasets, we applied our model to datasets collected from the primary motor cortex of two non-human primates (rhesus macaques) performing a reaching task (25). Neural recordings from these same individuals have been used in recent studies of deep representation learning (11), interpretable generative modeling (7), and adversarial domain adaptation (45).

**Motor cortex reaching datasets.** We use reaching as a simplified laboratory task to test our hypothesis that the *what* and *how* of movements could be disentangled. We consider spike sorted datasets from two rhesus macaques, Chewie and Mihi, both trained to perform a reaching task towards one of eight different directions after a cue. The reaching task has two different settings: Chewie performs the reaching task immediately after seeing the target on the screen (no waiting), while Mihi performs the reaching task with a waiting period of time (between 500-1500 ms) after receiving an auditory 'Go' cue. While carrying out these movement tasks, neural activities in primary motor cortex (M1) were recorded from both individuals. In these examples, the activity of a population of roughly one hundred single neurons was binned into 100ms intervals to generate approximately 1.3k data points per dataset. For each target direction, there are multiple trials/repeats. For each trial, the first 9 binned time points are selected for temporal decoding. For each individual, two days of neural recordings are considered, where different groups of neurons are recorded on different days.

**Experimental setup.** In our model, we apply a combination of two augmentations: (i) *spatial augmentations*, where neurons are randomly dropped (masked with zeros) from the input (with $p = 0.6$), and (ii) *temporal augmentations* that select a nearby point in time (randomly in a window,

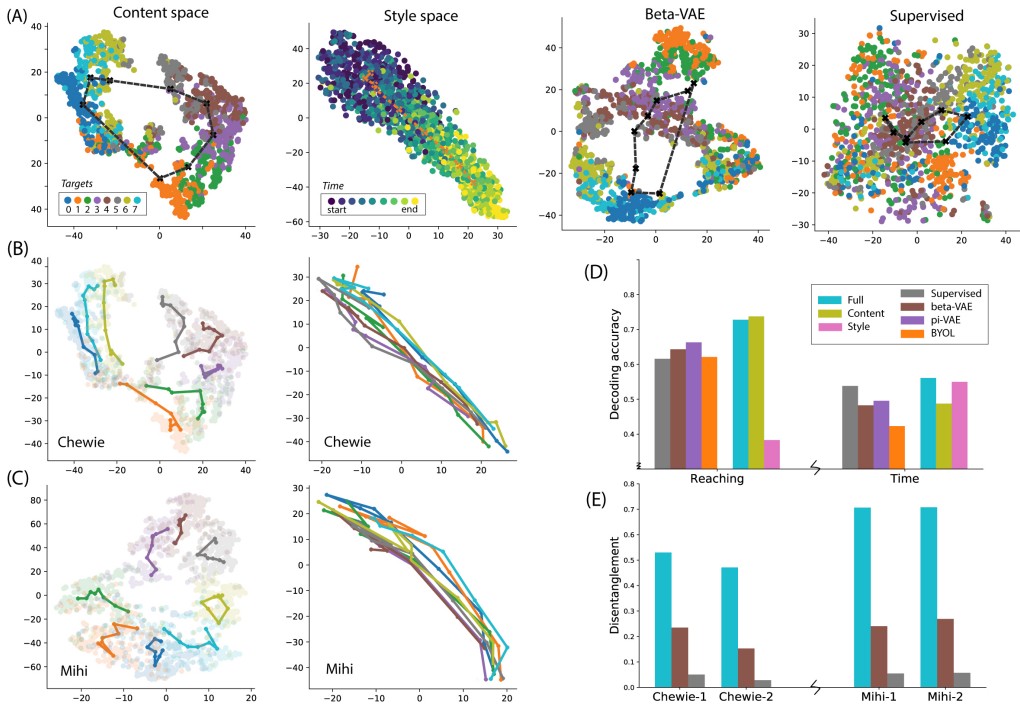

Figure 3: *Disentangling neural representations of movement in the primate motor cortex.* In (A), we show the representations formed by our model's content and style space when compared with the beta-VAE and a supervised network trained to decode reach direction. All of the visualizations are obtained after embedding the representations into 2D using tSNE. Below, we decompose the content and style space further by averaging over all trials towards a specific reach and visualizing their trial-averaged trajectory for Chewie (Day 1) in (B) and Mihi (Day 2) in (C). In (D), we compare the decoding accuracies over both reach direction and time for Chewie-1 for our Full, Content, and Style spaces, the benchmark models, and supervised decoders trained on either task. In (E), we show the results of our disentanglement score on all four reaching datasets. In this case, we compare the disentanglement over all of our latent space (Full) with the beta-VAE and supervised model trained on reach direction.

$\pm 5$ samples from target sample) as a positive example. All models have a 128-dim latent space, which is split for our model between into 64-dim blocks for the style and content space. All models are trained using one Nvidia Titan RTX GPU for 200 epochs using the Adam optimizer with a learning rate of 0.0005 (further details can be found in Appendix A). With $d$ as the total number of neurons, all generative models have an encoder and a symmetric decoder, where the encoder has three fully connected layers with size $[d, 128, 128]$, batch normalization, and ReLU activations. All discriminative models have an encoder of 4 linear layers with size $[d, 128, 128, 128]$, which was determined to be preferable after extensive hyperparameter optimization. In our experiments, we split the dataset into 80% for train and 20% for test.

**Investigating disentanglement in neural representations of movement.** After training our model, we examined the latent space structure by applying tSNE (46) to the Full space (considering both Content and Style jointly), as well as the Content and Style spaces individually. Chewie-1 and Mihi-2 are visualized in Figure 3, which admitted the highest reach decoding accuracy and most disentangled representations, respectively; visualizations of the remaining datasets can be found in Appendix C.1 where similar trends hold. When compared with a beta-VAE and Supervised decoder trained on the reach direction task, we observe that the Content space in our model and the beta-VAE have similar overall structure, with our model providing further separability and preservation of the task structure (circular positioning of targets, which is not intentionally encouraged by the loss function). The Style space provides a good embedding of the entire dataset along an axis where reach direction has been collapse but time is nicely organized. These results suggested that our model is good at separating semantic structure *without any labels* while also preserving the overall structure of the behavior.

To understand how much information our model extracts relevant to the two different downstream tasks, we explored the decoding accuracies in our reach and temporal decoding tasks on Chewie-1[1] (see Figure 3D, Appendix C.3). We examined the Full, Content, and Style spaces for our model on both tasks, and compared with the benchmark models and two separate supervised models trained on two tasks as the upper bounds. These measures provided further evidence of disentanglement: our Content spaces provide good decoding accuracies on reach decoding, while the Style space has little predictive power over reach direction (as anticipated). The reverse is true for the temporal decoding in the Content space. These results are promising indicators that disentanglement is indeed possible with our approach and that these decoding measures capture what we observe in our visualization.

We next measured the multi-task disentanglement scores across all four datasets. When examining the separability of our latent space across the two individuals (see Figure 3E), we found that the disentanglement (i.e., separation between the reach direction and the dynamics of the movement) quantified by the multi-task covariance scores for Chewie is on average lower than for Mihi in both cases; this observation may be interesting given that Mihi needs to wait before making a reach and has to delay their movement at the beginning. While the results of this analysis need to be studied further, this result provides initial evidence that our unsupervised method for disentanglement provides a useful lens into the distinction between the neural representation of different movement tasks.

**Quality of representations as measured through linear readouts.** Next, we conducted a comprehensive evaluation of the decoding of the reach direction (Table 1) and the temporal dynamics (Table 2). In this case, we compared our model with a supervised decoder trained on either task (Sup-Reach for reach decoding, Sup-Time for temporal decoding), a supervised (pi-VAE (7)), and an unsupervised (beta-VAE (23; 24)) generative approach for disentanglement (pi-VAE (7) and $\beta$-VAE (23; 24)), and two self-supervised methods for general representation learning (BYOL (47) and MYOW (11))[2]. These models provide a comprehensive collection of both supervised and unsupervised competitors for our tasks of interest.

The decoding accuracy analysis yielded the following interesting observations. First, we find that our model performs on par with or better than all the benchmark models on the reach direction decoding task. Second, we find that our model performs comparably with beta-VAE and the supervised model (Sup-Time) on the time decoding task, while other self-supervised models fail to capture the temporal structure that they have built invariance to. These results indicate that both *Swap-VAE* and MYOW, two self-supervised approaches, outperform supervised decoders on reaching direction classification, due to the strong regularization from their alignment losses.

The power of our model shines in its ability to extract both the reaching target as well as the temporal structure from the data. As we show in both Figure 3D and Table 2, using a SSL-based alignment loss (BYOL, MYOW, and *Swap-VAE*'s Content space) with temporal jitter augmentations builds temporal invariance, which makes it challenging to decode temporal structure of the reaches. On the other hand, *Swap-VAE* introduces an additional set of latent variables (Style space) that carry additional information and make it possible to achieve high accuracy in both reach and temporal decoding.

**Testing the generative quality of the model.** A key component of *Swap-VAE* is the integration of SSL with a generative modeling framework. Thus, it is important to gauge the quality of the generated data. As shown in Figure 4, the direct reconstruction of the neuron firing rate is realistic in terms of both the class-conditioned firing rates, and the dynamics of individual neurons. When we analyzed the RMSE of the fitted rate for all neurons in our model against those from a VAE, we found our model has a lower error and could reconstruct data more faithfully than the VAE. Our results suggest that Swap-VAE provides a good denoised estimate of neural activity that captures information related to the behavior (rather than factors extraneous to the movement).

To further demonstrate that our generated neuron activities are useful for downstream tasks, we use our trained model to generate new samples and mix them with original training samples when training a supervised classifier. All supervised models are trained for 400 epochs, with the same model settings as mentioned in the experimental setup. We tested on one dataset from each individual (Chewie-1,

---

[1] Chewie-1 yields the highest reach decoding accuracy of all datasets, which motivated this choice.

[2] We include the reported numbers for reach decoding, and also reproduced MYOW for temporal decoding (see reproduced results in Appendix B.1). With changes to the network architecture and set of augmentation operations, MYOW recently reported even higher accuracies on reach decoding.

Table 1: *Accuracy (in %) for reach direction classification on neural datasets.* See Appendix B for the standard deviation of the benchmark models.

| | | Sup-Reach | pi-VAE | beta-VAE | BYOL | MYOW(11) | Ours |
|---|---|---|---|---|---|---|---|
| Chewie-1 | acc | 61.59 | 66.30 | 64.34 | 62.12 | 70.41 | **72.81**(±1.40) |
| | $\delta$-acc | 77.58 | 82.93 | 80.83 | 81.27 | **86.24** | 85.04(±0.94) |
| Chewie-2 | acc | **69.71** | 61.33 | 60.24 | 57.25 | 60.95 | 68.97(±3.41) |
| | $\delta$-acc | 78.18 | 73.63 | 80.09 | 76.89 | 81.36 | **83.66**(±1.81) |
| Mihi-1 | acc | 62.86 | 62.63 | 58.11 | 60.03 | **70.48** | 64.26(±0.89) |
| | $\delta$-acc | 79.10 | 79.20 | 75.98 | 78.82 | **83.24** | 82.18(±1.55) |
| Mihi-2 | acc | 60.72 | 62.70 | 60.23 | 59.94 | 64.35 | **66.12**(±0.87) |
| | $\delta$-acc | 74.02 | 76.89 | 77.89 | 78.10 | 80.58 | **82.74**(±0.77) |

Table 2: *Accuracy (in %) for temporal decoding classification on neural datasets.* See Appendix B for the standard deviation of the benchmark models.

| | Sup-Time | pi-VAE | beta-VAE | BYOL | MYOW | Ours |
|---|---|---|---|---|---|---|
| Chewie-1 | 53.80 | 49.56 | 48.24 | 42.30 | 22.45 | **56.09**(±1.80) |
| Chewie-2 | 55.58 | 54.45 | **63.76** | 39.69 | 31.96 | 62.54(±2.02) |
| Mihi-1 | 55.93 | 52.66 | 54.10 | 43.46 | 21.88 | **57.77**(±3.30) |
| Mihi-2 | **58.75** | 48.03 | 47.89 | 38.16 | 22.09 | 49.65(±2.01) |

Mihi-2) and computed the improvement in accuracy as we increase the number of generated samples included in the training set (50%, 100%, 200%). In all cases, we found some improvement in the accuracy of the model, with roughly 5% and 3.5% gains over the supervised baseline in Chewie and Mihi, respectively. We note that the supervised models trained with generated samples still does not surpass the *Swap-VAE* (details in Appendix D).

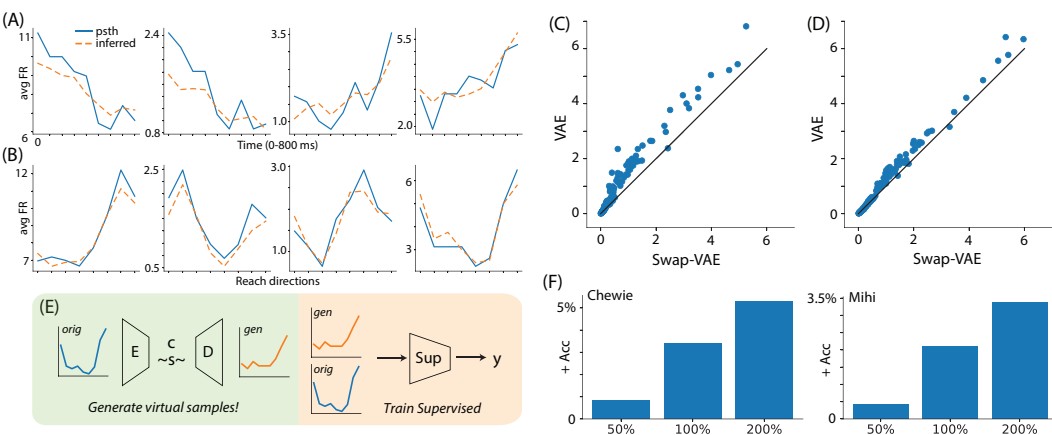

Figure 4: *Testing Swap-VAE's ability to reconstruct and generate new neural activity.* Reconstruction of the firing rates from example neurons over time (A) and across different reaching directions in (B). We further validate the reconstruction accuracy of our model by comparing the RMSE obtained with our model (x-axis) vs. the RMSE for the VAE (y-axis) for different reaching directions (C) and over time (D). (E) shows a sketch of how we generate virtual samples (green) and use them to train a supervised classifier (orange). (F) highlights the improvement in classification accuracy as we increase the amount of generated data fed into a supervised decoder (left, Chewie-1; right, Mihi-2.)

## 4.3 Model ablations: Testing our *BlockSwap* augmentation

We studied different variants of the proposed loss functions in Equations (2) and (3) and how different data augmentation operations impact decoding accuracy (Table 3). The three variants of the loss are the following: (i) when only removing the alignment term in Eq. 2 (no L2), (ii) when only removing the *BlockSwap* augmentation (No-Swap), and (iii) when removing the alignment term and the original

Table 3: *Model ablations.* Accuracy (in %) of different variants of our proposed model when tested on Chewie-1.

|  | no L2 | no-swap | swap-only | vanilla VAE | S-Aug | T-Aug | Ours |
|---|---|---|---|---|---|---|---|
| acc | 71.63 | 63.17 | 68.36 | 63.79 | 66.77 | 71.15 | **72.81** |
| delta-acc | **85.62** | 83.20 | 83.02 | 79.10 | 81.03 | 83.34 | 85.04 |

reconstruction term (Swap-only). We also report the results obtained with a vanilla VAE, while beta-VAE results are in Table 1. Ablations of different dimensions of the content and style space are included in Appendix E.

In our experiments, we find that almost all of the variants of our decomposed loss functions performs better than a vanilla VAE or a beta-VAE. Adding the *BlockSwap* loss term improves performance overall, with our highest decoding accuracies being obtained with this model. When we use the content swapping technique, we can remove the L2 alignment loss with minimal change in performance, but in general, including this alignment terms provides an additional parameter to give more flexible control. This shows that our proposed model is stable and that our proposed swapping loss provides a strong boost in performance.

We tested the spatial-only (S-Aug) and temporal-only (T-Aug) conditions separately in Table 3. In this case, we can see that they all perform reasonably well, although they are both worse than our final model where we combine both spatial and temporal augmentations. As we know, the selection of the data augmentations is critical for the performance of a representation learning model (32; 48; 49). Our model needs even fewer data augmentation operations than MYOW to achieve a good performance, highlighting the power of our approach.

## 5  Conclusion

This paper introduces a new self-supervised approach, *Swap-VAE*, for generative modeling of neural activity. Our proposed method leverages self-supervised alignment to decompose neural activity into different latent subspaces, with the goal of provide insights into the relationship between neural activity and animal behaviour.

While our model provides a good balance between both reach and temporal decoding using simple augmentations (dropout, temporal jitter), we note that, just like other self-supervised learning methods, the performance of our method relies on our choice of augmentations, which might be dependent on the downstream task of interest. Exploring how different augmentations affect performance on different downstream tasks may yield insights into how information is coded in the brain, which will be an exciting line of future research.

In our current formulation, we only consider dropout and localized temporal shifts as augmentations of the neural activities. However, other works like (11) use a nearest-neighbor approach to link brain states that are temporally nonlocal or may span different trials. Through combining our approach with this nonlocal view mining strategy, we may be able to build even further invariance into our model's content space. Combining our SSL-backed approach with a sequential encoder (10) or transformer (50) is another exciting line of future research that can be used to model latent structure over longer timescales.

Beyond demonstrating the effectiveness of our method, our analysis of neural activity patterns facilitated by *Swap-VAE* had other intriguing outcomes. There is an active debate in contemporary neuroscience about the role of preparation in the neural activity associated with movement (51; 52; 53), with some evidence suggesting that preparation makes the neural activity driving movement more stable and robust. As we point out in Section 4.2, Chewie and Mihi were instructed to make reaches differently, with Mihi required to wait for an auditory cue before performing each movement. We found that the disentanglement in Mihi was more pronounced than Chewie, both in terms of decoding accuracy across content and style spaces and multi-task covariance scores. It is possible that this is due to a weakness of our method, which could perhaps be resolved by tuning hyperparameters, increasing the size of the network, or using some new augmentations. However, it might be possible that preparation leads to more disentangled neural activity and that this is discernible in the evaluation of the multi-task disentanglement score, which should be explored carefully across more individuals.

## Acknowledgements

This project was supported by NIH award 1R01EB029852-01 (RL, MA, CHL, KBH, ELD), 1R01NS118442-01 (KBH), NSF awards IIS-1755871 (ELD) and IIS-2039741 (ELD), a NSF GRFP (MD), as well as generous gifts from the Alfred Sloan Foundation (RL, ELD) and the McKnight Foundation (RL, MA, CHL, ELD).

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
