# Appendix

## A  Implementation details

### A.1  Model implementation details

The *Swap-VAE* consists of an encoder and a symmetric decoder. Denote $d$ as the total number of neurons and $k$ as the latent space dimension. The encoder contains three linear layers with output size $[d, k, k]$, each but the last layer is followed by batch normalization, with $\mathrm{eps} = 0.00005$ and $\mathrm{momentum} = 0.1$, and the ReLU activation. The decoder contains three linear layers with output size $[k, k, d]$ where each but the last layer contains a Batch normlization and the ReLu activation similar as above. The last layer of the decoder is followed by a SoftPlus activation with $\mathrm{beta} = 1$ and $\mathrm{threshold} = 20$. For synthetic experiments, all models are trained using a Nvidia Titan RTX GPU for 100,000 iterations using an Adam optimizer with a learning rate of 0.0001. We used a batch size of 256 for both synthetic experiments and monkey reaching experiments.

Following the standard linear evaluation procedure in self-supervised learning works (32; 34), we used an one linear layer network as the linear decoder for the decoding accuracy. For the linear decoder, we used the Adam optimizer with learning rate 0.005 and weight decay 0.00001 throughout, for both reach and temporal decoding tasks. The hyperparameters are decided based on a grid search of learning rate and weight decay on the supervised pre-trained feature space for reach directions.

### A.2  *BlockSwap* implementation details

Our *BlockSwap* latent augmentation operation is implemented as shown in Fig S1. Here we use the colors blue and green to represent two 'styles', and the colors pink and purple to represent the same 'content' that is shared by two augmented views.

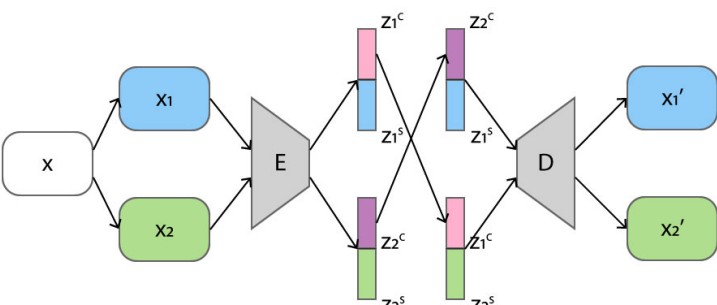

Figure S1: *BlockSwap latent space augmentation.* The model separates the latent space into two parts and solves an instance-specific alignment over time and neurons in the first part of the space, while building good reconstructions at the output of the decoder using both parts of the latent space.

### A.3  Dataset details

**Data collection.**  We used the neural activity dataset that is collected from two rhesus macaque monkeys (Chewie and Mihi). They were trained to move the computer cursor to reach a target on a screen. Chewie is instructed to move the cursor from the center of the screen (wait for 500-1500 ms) to the target (in 1000-1300 ms) immediately after the target appears. Mihi, differently, is instructed to wait another 500-1500 ms to plan the movement after the target appears but before an auditory 'Go' cue. Their spiking neural activities on primary motor cortex (M1) were recorded by surgically implanted electrode arrays. The neural activities were thresholded and sorted during the data collection stage.

**Data pre-processing.**  For the monkey reaching dataset, after the data is binned, we compute the variance of each neurons and remove the static neurons with zero variance. This procedure results in a total numbers of neurons of 163 for Chewie day1, 148 for Chewie day 2, 163 for Mihi day 1, and 152 for Mihi day 2.

# B Comparing performance across multiple initializations

In this section, we first include the standard deviation of the benchmark models after 5 repeats, and then we investigate the variance of *Swap-VAE* in details.

## B.1 Stability of the benchmark models

In table S1 and table S2, we include the standard deviation of the benchmark models that is computed based on 5 random seeds. The model initial weights, the model training, and the linear evaluation layer are all randomized differently based on the random seed. Here, the MYOW results are from the reproduced version based on the hyperparameters, augmentations, and optimization details that we selected for BYOL.

Table S1: *Accuracy (in %) for the benchmark models in reach direction classification task.* The standard deviation is computed over 5 random initializations.

| | | Supervised | pi-VAE | beta-VAE | BYOL | MYOW |
|---|---|---|---|---|---|---|
| Chewie-1 | acc | 61.59(±2.04) | 66.30(±1.30) | 64.34(±1.08) | 62.12(±2.27) | 67.61(±1.41) |
| | $\delta$-acc | 77.58(±1.28) | 82.93(±1.34) | 80.83(±1.08) | 81.27(±1.08) | 81.63(±2.18) |
| Chewie-2 | acc | 69.71(±1.85) | 61.33(±0.86) | 60.24(±3.18) | 57.25(±1.11) | 63.06(±3.68) |
| | $\delta$-acc | 78.18(±2.96) | 73.63(±1.80) | 80.09(±1.75) | 76.89(±1.30) | 81.46(±3.28) |
| Mihi-1 | acc | 62.86(±1.08) | 62.63(±1.42) | 58.11(±1.52) | 60.03(±0.95) | 64.94(±2.24) |
| | $\delta$-acc | 79.10(±1.20) | 79.20(±2.30) | 75.98(±1.10) | 78.82(±1.68) | 82.29(±3.94) |
| Mihi-2 | acc | 60.72(±1.74) | 62.70(±0.90) | 60.23(±0.96) | 59.94(±1.39) | 57.30(±1.51) |
| | $\delta$-acc | 74.02(±3.18) | 76.89(±1.28) | 77.89(±1.18) | 78.10(±1.85) | 71.80(±1.48) |

Table S2: *Accuracy (in %) for the benchmark models in temporal decoding classification.* The standard deviation is computed over 5 random initializations.

| | Sup-Time | pi-VAE | beta-VAE | BYOL | MYOW |
|---|---|---|---|---|---|
| Chewie-1 | 53.80(±1.68) | 49.56(±1.33) | 48.24(±1.20) | 42.30(±0.98) | 22.45(±1.62) |
| Chewie-2 | 55.58(±1.52) | 54.45(±1.83) | 63.76(±1.84) | 39.69(±1.15) | 31.96(±3.28) |
| Mihi-1 | 55.93(±2.09) | 52.66(±2.10) | 54.10(±0.98) | 43.46(±0.87) | 21.88(±2.24) |
| Mihi-2 | 58.75(±0.39) | 48.03(±1.21) | 47.89(±1.37) | 38.16(±1.22) | 22.09(±1.27) |

## B.2 Stability of the SwapVAE

To examine the robustness and stability of our model, we run experiments to study the performance of *Swap-VAE* over different random initializations (see Table S3). We report the mean and the standard deviation of the model accuracy in two ways, where 'Whole' denotes the result we obtain when we train and evaluate the whole networks with 5 different random seeds, and in 'Evaluation' we further select the model with the best performance from the previous five models and then retrain the linear decoding layer training using 5 random initializations. Our results confirm that Swap-VAE maintains a gap over other methods and provides insights into the different sources of variance in our model.

Table S3: *Accuracy and standard deviation (in %) for reach direction classification on neural datasets.*

| | | Whole | | Evaluation | |
|---|---|---|---|---|---|
| | | Mean | SD | Mean | SD |
| Chewie-1 | acc | 72.81 | 1.40 | 74.47 | 0.27 |
| | $\delta$-acc | 85.04 | 0.94 | 86.25 | 0.26 |
| Chewie-2 | acc | 68.97 | 3.41 | 75.37 | 0.62 |
| | $\delta$-acc | 83.66 | 1.81 | 86.12 | 0.29 |
| Mihi-1 | acc | 64.26 | 0.89 | 66.00 | 0.71 |
| | $\delta$-acc | 82.18 | 1.55 | 83.31 | 0.66 |
| Mihi-2 | acc | 66.12 | 0.87 | 66.94 | 0.58 |
| | $\delta$-acc | 82.74 | 0.77 | 84.24 | 0.11 |

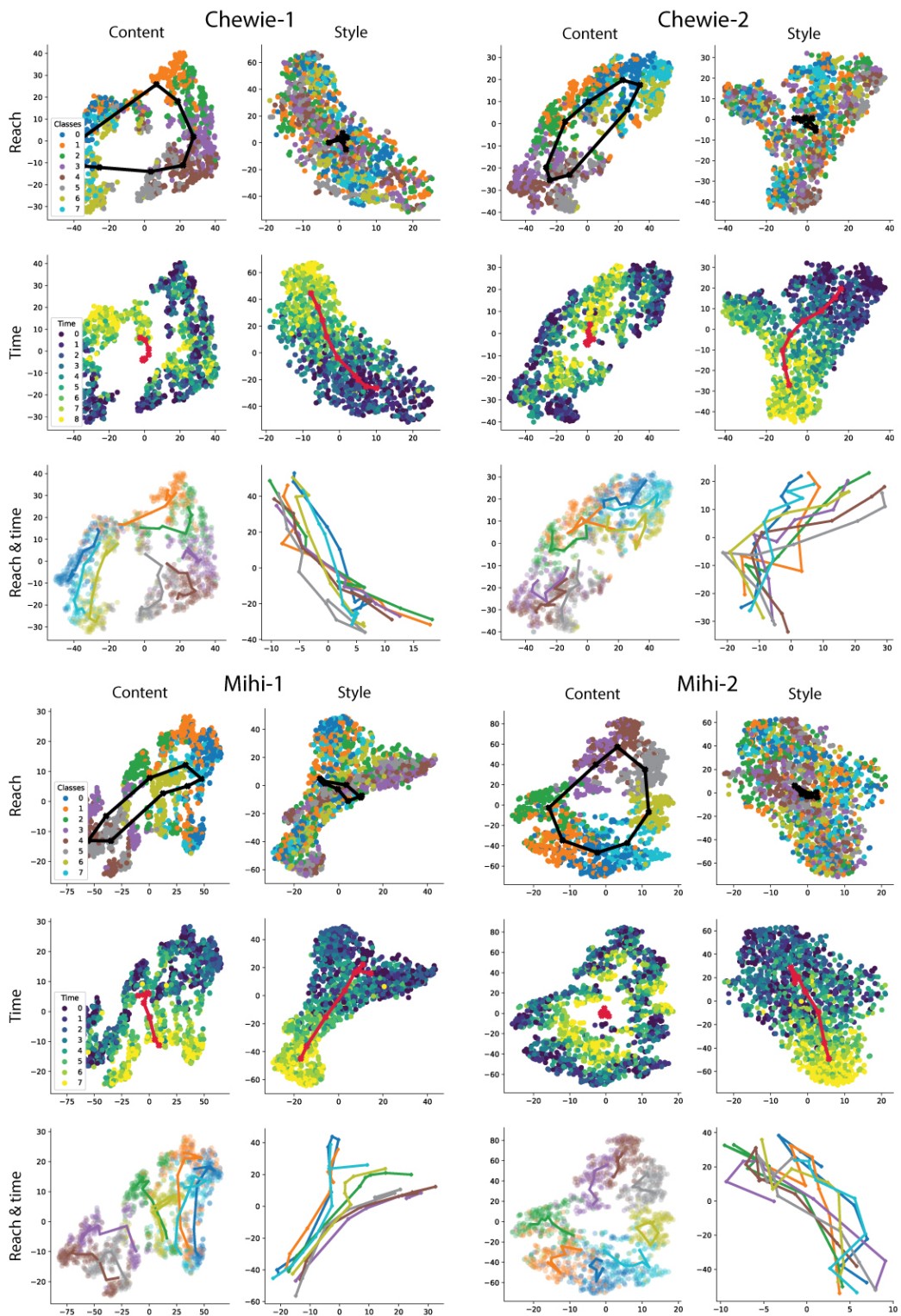

Figure S2: *Visualization of the representations learned by Swap-VAE.* For each dataset, the reach directions, the dynamics, and the reach directions with the dynamics for each classes are plotted after applying tSNE to embed the data.

## C   Additional latent space quality experiments

### C.1   Latent space visualizations

In Fig S2, we show the *Swap-VAE* latent subspaces visualizations for Chewie and Mihi. Similar to Fig 3, we plot the latent space structure by applying tSNE to the Content and Style spaces individually for all four datasets. For each dataset, the content space is shown on the left side while the style space is shown on the right side. For each dataset, the first row is colored based on the reach directions, the second row is colored based on the dynamics, and the third row is colored based on the reach directions with the average 'trajectory' of each reach direction represented.

### C.2   Time decoding details

The time decoding accuracy is computed similarly as the reach direction accuracy, but in this case, the supervised model is trained to predict how far into each reach each sample is. The first bin always marks the beginning of a reach and each reach is restricted to its first 9 bins, resulting in a classification problem with 9 classes.

### C.3   Multi-task decoding scores

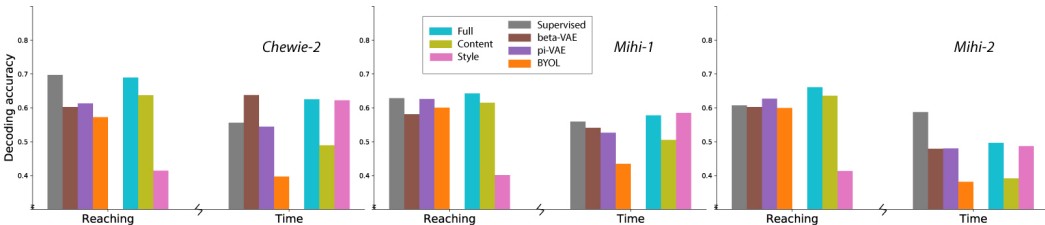

Figure S3: *Visualizations of reach and temporal decoding accuracy.* Each plot contains the reach and time decoding for a different neural recording, where we compare the decoding from either the Full, Content, or Style space of our model with other baseline methods. For the supervised decoder, we train two different decoders on either the reaching or temporal decoding task.

## D   Additional details on the generative quality experiment

We describe the details on the generative quality experiment (Section 4.2) here. With a pre-trained and fixed *Swap-VAE*, we generate datapoints for data augmentation as below: We compute the latent representation of an existing datapoint, and split it to a 'content' latent and a 'style' latent as defined by the model. To generate a new datapoint, the 'content' latent is fixed and the 'style' latent is varied with added noise. In practice, the noise is randomly sampled from $0.2 \times \mathcal{N}(\mathbf{0}, \mathbf{I})$. A new latent vector is constructed by concatenating the original 'content' vector with the new 'style' vector, and is then passed into the decoder to generate new instance-specific neural activities.

While the generated neural activities can effectively improve existing supervised models when the generative models themselves surpass the supervised models (Chewie-1 and Mihi-2), they do not improve existing supervised models when the generative models have similar or worse performance as the supervised models (Chewie-2 and Mihi-1). It is reasonable to surmise that the performance of the generative model is the upper bound of the supervised model trained via generated neural activities. However, we note that the generated neural activities are still valuable for other downstream tasks.

## E   Ablation experiments

### E.1   Different latent dimensions

We conducted two types of ablation experiments that tackle two questions:

- What would happen if we split the 128-dim latent space unevenly between content and style, and how does the proportion effect the performance?

- What would happen if we allocate 128-dim to the 'Content' latent space, which equals the latent space dimension of other models, and have additional 'Style' latent dimensions?

**Dimension of content vs. style variables.** In Table S4, we fix the Full latent space dimension to be 128-dim, and vary the size of the Style latent space dimension. From the table, we observe that our models have comparably good performance when splitting the latent variables into different configurations. Note that when $s = 128$, the proposed model would converge to a beta-VAE model (see Table 1 for beta-VAE performances).

Table S4: *Accuracy (in %) for different latent space separation.*

|          |           | $s = 0$ | $s = 16$ | $s = 32$ | $s = 64$ | $s = 96$ |
|----------|-----------|---------|----------|----------|----------|----------|
| Chewie-1 | acc       | 62.56   | 68.85    | 69.65    | **73.44** | 71.22   |
|          | delta-acc | 79.23   | 82.27    | 83.63    | **85.38** | 82.63   |
| Chewie-2 | acc       | 59.80   | 65.56    | 65.75    | 66.06    | **67.23** |
|          | delta-acc | 77.34   | 82.55    | **84.14** | 82.26   | 83.50   |
| Mihi-1   | acc       | 62.92   | **65.60** | 63.62    | 65.15    | 64.20   |
|          | delta-acc | 79.39   | 79.56    | 80.28    | **81.16** | 80.49   |
| Mihi-2   | acc       | 64.10   | 65.15    | 63.25    | **67.78** | 63.28   |
|          | delta-acc | 80.46   | 81.55    | 79.75    | **84.05** | 79.90   |

**Testing latent spaces of different dimensionality.** Here, we fix the Content space dimension to be 128-dim, and add additional Style latent space dimensions. In Table S5, we report the accuracy under linear evaluation on the Content latent space only. This comparison is reasonable since only the 128-dim Content space is regularized by the alignment loss to classify the reach directions. This result shows that our model has the possibility to achieve even better performances when the Content space has the equal dimension as other models.

Table S5: *Accuracy (in %) for different additional Style latent space dimensions.*

|          |           | s=32   | s=64    | s=128  |
|----------|-----------|--------|---------|--------|
| Chewie-1 | acc       | 72.82  | **74.13** | 69.79 |
|          | delta-acc | 83.39  | 84.74   | **85.47** |
| Chewie-2 | acc       | 65.09  | **68.42** | 67.47 |
|          | delta-acc | 82.03  | 82.88   | **86.14** |
| Mihi-1   | acc       | 65.82  | **68.90** | 62.32 |
|          | delta-acc | 82.29  | **84.88** | 78.44 |
| Mihi-2   | acc       | **66.52** | 65.03 | 65.67 |
|          | delta-acc | 80.94  | **82.22** | 81.80 |

We also note that the accuracy evaluated using the Full latent space features is in general similar or slightly better than the presented values.

## E.2 Disentanglement score for different ablation models

One interesting question to ask is that: does *BlockSwap* facilitate disentanglement? In order to answer this question, we provide the disentanglement score for all of the loss function ablations that we tested in table S6. Our results suggest that keeping the alignment loss and the original reconstruction term but removing the *BlockSwap* augmentation (No-Swap) does not facilitate better disentanglement compared with beta-VAE. Removing the alignment term but including the *BlockSwap* (no L2) provides better disentanglement. Removing the alignment term and the original reconstruction term (Swap-only) provides the closest disentanglement score as our model. *Swap-VAE* provides the best disentanglement performance.

Table S6: *Multi-task disentanglement score for different ablation models.*

| beta-VAE | no L2 | No-Swap | Swap-only | Ours |
|----------|-------|---------|-----------|------|
| 0.2070 | 0.3455 | 0.2089 | 0.4914 | 0.5380 |

### E.3 More details about data augmentation operations

The model we presented in the paper used only used Dropout (S-Aug) and Temporal shift (T-Aug) as two data augmentation operations. We are aware that more possible data augmentation operations exist, e.g. the Pepper operation and Noise operation. However, when we tried to add those data augmentations into our model, the model performance did not improve further. It is reasonable to surmise that generative models may not need data augmentation operations that are as intense as that in recent self-supervised learning works.