# OpenReview forum: "Drop, Swap, and Generate: A Self-Supervised Approach for Generating Neural Activity"
_NeurIPS.cc/2021/Conference — NeurIPS 2021 Oral_

### Official Review · Reviewer_r6nv · 2021-07-14

**Rating:** 7
**Confidence:** 4

**Summary:**

The authors propose a self-supervised approach to learning representations of neural population activity in motor cortex. The approach is based on disentangling slowly varying action classes (content) from temporal dynamics of the action (style). They show that their approach facilitates decoding of eight action classes (reach directions) and compare with previous methods based on autoencoding (beta-VAE) or contrastive learning (MYOW). Overall the approach seems sound and promising. Unfortunately, it is difficult to verify the claims of the paper, since the analyses are very selective and not rigorous.

While I am generally supportive of the paper, its current issues let me lean towards rejection unless the problems will be resolved so the claims can be assessed. My suggested improvements could potentially be done during the author response phase and I'm happy to revise my score if the authors provide solid evidence for their claims.

### Update after discussion period
The authors have thoroughly addressed my concerns and performed additional experiments. I am confident that after the promised revisions the paper will be very worthwhile publishing at NeurIPS and changed my score to 7 accordingly.

**Limitations And Societal Impact:**

Limitations could be addressed more openly, e.g. the limits on disentanglement w.r.t. decoding time.

**Main Review:**

## Strengths
+ Simple and novel self-supervised representation learning method
+ Potentially improves decoding of task-relevant variables


## Weaknesses
- Not clear if claims are fully supported by the data
- Possible selective reporting/cherrypicking of results
- Writing not always clear


## Detailed comments on weaknesses

### Selective reporting

Unfortunately, the data analysis is not done very rigorously (no statistics at all) and on subsets of the data. Also, I can't help but getting the impression that some results that don't fit as nicely into the story as others are conveniently moved to the supplement.

(a) The authors perform the analyses shown in Fig. 3D and Table 2 only on the Chewie-1 dataset. Given the substantial variation across datasets shown in Fig. S2, Table 1 and Table S1, this is a problem – especially since it's not stated when and why this dataset was selected.

Action items: All analysis that are used to draw conclusions should be done on all datasets and a measure of uncertainty (e.g. standard error of the mean/confidence intervals across datasets) should be reported, so the reader can assess to what extent the reported results can be attributed to randomness in the data vs. the method proposed.

(b) The reader would get a better impression of the reliability of the method (or the variability across datasets) if the examples in Fig. 3B+C were not cherry-picked to show the circular arrangement of the latent space, but rather show the datasets of the first days for both monkeys. This is also worrisome as the authors make a claim about the method identifying the circular arrangement in the latent space earlier in Fig. 2 and it is not clear to me what aspect of the loss function/training procedure would encourage such behavior (see below for a comment on this point).

Action item: show more representative data in Fig. 3 B+C.

(c) In the section on decoding, the authors show only Table 1, the accuracy for reach direction, to make the point that their method is superior to earlier approaches. However, Table S1, which shows that the method does not lead to an improvement in decoding time, is relayed to the appendix without even stating the results in the main text. Moreover, Table S2 is lacking the stronger baselines, so presumably the proposed method would look even worse in this respect if this table was complete.

Action items: Table S1 should be complete with all baselines, moved into the main text and the paper's claims be adjusted to be consistent with the full set of results. If space is a problem, the authors could summarize reach direction and time decoding in one table by showing only the averages across datasets in the main paper and providing the full results per dataset in the appendix.


### Further comments (action required)

- Line 311ff. The difference between (i) and (ii) is not clear to me. Could you clarify what removing "the original reconstruction term" means? If there is no reconstruction term, how is the decoder learning anything? Or is it meant such that (i) includes both reconstruction terms from Eq. (3) while (ii) reconstructs only the content-swapped sample?

- Multi-task disentanglement score (line 158ff.). This section needs a substantial revision. I did not understand what's happening here, despite several reading attempts. The variables y_c and y_s are not defined anywhere, it's not clear to me how they relate to z_i and what variances are computed in lines 166–168.

- Fig. 3D: bars should start at zero, not at 0.3

- Fig. 4D: The y axis is labeled "VAE". Is this the same beta-VAE as everywhere else or a vanilla VAE (beta=1)? Given the selective analyis at other places, I'd like to make sure that this figure is not using a strawman as baseline.


### Minor comments (action recommended, but not required)

- The meaning of content vs. style could be discussed a bit better. I am not sure if the "style" is a well-defined concept here. The authors seem to equate it with temporal dynamics, but the temporal dynamics are clearly also encoded in the content space (as can be seen quite well in Fig. S2). So while the content space enables good decoding of the reach direction, time can be decoded from both subspaces. In that sense, the disentangling isn't very clear.

- Evaluation metric for decoding: Why turning it into classification if you treat it as regression? Instead of applying two different sets of boundaries for acc and delta-acc, why not just report mean squared angular error as a metric of reconstruction accuracy?

- Fig. 2: I understand that the four classes lie on a circle, but it's not clear to me why this arrangement should be retained in the latent space. The authors seem to think of this as important/desirable and their method retains the arrangement while the beta-VAE doesn't. However, it's not clear to me what aspect of the loss function/training procedure would encourage such behavior and whether it's not just a random effect of initialization. If I'm not mistaken, if you pick four locations at random, it should be 50:50 whether they maintain the circular arrangement or not.

- Line 296 "rather than noise": I am not sure if noise is the right term here. Presumably a substantial fraction of neuronal variability is due to variability in the exact arm trajectory across trials, which isn't really noise.

- Line 297ff. I am not sure what's the point of this paragraph. As the authors state themselves at the end, if you want to decode, you're better off just using the SwapVAE latent than training a supervised classifier from generated samples. I think this paragraph can be removed to make space for more important points and showing some results that are currently hidden in the appendix (see above).

- Line 267ff. This paragraph seems like pure speculation. Given the amount of variability across datasets and the fact that it's just a single animal, this results could be due to many things, the delay period being only one of them. I think the authors would do themselves a favor by not jumping to a premature conclusion here. So again, this paragraph could be cut and replaced by more important material.

**Time Spent Reviewing:**

4

---

> ### Author Response · Authors · 2021-08-10
> **Response to Reviewer r6nv**
>
> Thank you for your review and critical examination of the work. To try to address your concerns, we have carried out multiple experiments and report the new results below. In addition to these new experiments, we plan to address your concerns regarding the claims made in the paper and provide a more fair showing of results across many individuals. We hope you might re-evaluate your score based upon these additional results and our responses.
>
> ### **Main responses and action items**
>
> 1. **Comparing model performance across random initialization**
>
> * To address your concern about the stability of our approach and the need for statistics throughout, we ran two different experiments involving randomization and reported the mean and standard errors for our method and other competitors (supervised models, pi-VAE, beta-VAE, and BYOL). We report the accuracies for our model in the last two columns, where Swap-VAE (1-stage) denotes the result we obtain when we train and evaluate networks with 5 different random seeds, and in Swap-VAE (2-stage) we further select the model with the best performance and then retrain the linear decoding layer training using 5 random initializations. Our results confirm that Swap-VAE maintains a gap over other methods and provides insights into the different sources of variance in our model. These results will be provided in a revised manuscript.
>
> |         |   | Supervised | pi-VAE | $\beta$-VAE | BYOL (repro) | BYOL [1] | MYOW [1] | Ours (1) | Ours (2) |
> |---------|-------|-------| -------| -------| -------| -------| -------| -------|  -------|
> | Chewie-1 | acc | 61.59$\pm$2.04 | 66.30$\pm$1.30 |64.34$\pm$1.08 |62.12$\pm$2.27 | 63.80 | 70.41 | 72.81$\pm$1.40 | 74.47$\pm$0.27 |
> |  | $\delta$-acc | 77.58$\pm$1.28 | 82.93$\pm$1.34 |80.83$\pm$1.08 |81.27$\pm$1.08 |81.90 |86.24  |85.04$\pm$0.94 |86.25$\pm$0.26 |
> | Chewie-2 | acc | 69.71$\pm$1.85 | 61.33$\pm$0.86 |60.24$\pm$3.18 |57.25$\pm$1.11 | 57.17| 60.95 |68.97$\pm$3.41 |75.37$\pm$0.62 |
> |  | $\delta$-acc | 78.18$\pm$2.96 |73.63$\pm$1.80 |80.09$\pm$1.75 |76.89$\pm$1.30 | 77.36| 81.36 |83.66$\pm$1.81 |86.12$\pm$0.29 |
> | Mihi-1 | acc | 62.86$\pm$1.08 |62.63$\pm$1.42 |58.11$\pm$1.52 |60.03$\pm$0.95 | 59.50 | 70.48 |64.26$\pm$0.89 | 66.00$\pm$0.71 |
> |  | $\delta$-acc | 79.10$\pm$1.20 |79.20$\pm$2.30 |75.98$\pm$1.10 |78.82$\pm$1.68 |79.78 |83.24 |82.18$\pm$1.55 |83.31$\pm$0.66 |
> | Mihi-2 | acc | 60.72 $\pm$1.74 |62.70$\pm$0.90 |60.23$\pm$0.96 |59.94$\pm$1.39 |60.82 |64.35 |66.12$\pm$0.87 |66.94$\pm$0.58 |
> |  | $\delta$-acc | 74.02$\pm$3.18 |76.89$\pm$1.28 | 77.89$\pm$1.18 |78.10$\pm$1.85 | 78.30 | 80.58 | 82.74$\pm$0.77 | 84.24$\pm$0.11 |
>
> _Table 1: Reach decoding accuracy across different random initializations_
>
> 2. **Extended comparisons for temporal decoding**
>
> * In our first submission, we did not include the table with temporal decoding results in the main text because we believed that showing disentanglement (Figure 3D, 3E), reach decoding accuracy (Table 1), and examining the generative capabilities of the model (Figure 4) touched on all key aspects of the representation quality and disentanglement with our model. Based upon the reviewer’s suggestions, we will include reach and time decoding across multiple models in the main text (in Figure 3 where we provide decoding and disentanglement scores for all datasets and many models).
>
> * Before describing our expanded experiments to address your concerns, however, we would like to provide some insight into the logic behind our choices (which we hope to make more clear in the revised manuscript). Swap-VAE combines both SSL and generative modeling and thus, we wanted to build appropriate comparisons across these relevant dimensions. In the case of decoding the reach direction, where we use a self-supervised loss, we compared with other SSL approaches (MYOW and BYOL). In the case of the temporal decoding task, we compared with generative modeling approaches that try to capture the full variance in the data in order to reconstruct it (beta-VAE).
>
> * With all that said, based upon your suggestion, we reproduced both BYOL and pi-VAE models and re-trained the classifier to decode time. We expect BYOL and Swap-VAE content space to provide similar temporal decoding performance, since they are all regularized to maintain temporal invariance, while in our results, Swap-VAE content space performs better in general. The generative models do a better job of decoding time, with the Swap-VAE Style and Full spaces providing the best decoding of time. We believe that these results, combined with our quantification of disentanglement provide multiple dimensions over which we try to demonstrate the usefulness of our approach -- both in building disentangled representations and also for generating neural activity.
>
> |        | Supervised | pi-VAE | $\beta$-VAE | BYOL (repro) | Ours (Content) | Ours (Full) |
> |--------|-------| -------| -------| -------| ------- | -------|
> | Chewie-1 | 53.80$\pm$1.68 | 49.56$\pm$1.33 | 48.24$\pm$1.20 | 42.30$\pm$0.98 | 48.74$\pm$0.71 | 56.09$\pm$1.80 |
> | Chewie-2 | 55.58$\pm$1.52 |54.45$\pm$1.83 |63.76$\pm$1.84 |39.69$\pm$1.15 |48.94$\pm$1.21 |62.54$\pm$2.02 |
> | Mihi-1 | 55.93$\pm$2.09 |52.66$\pm$2.10 |54.10$\pm$0.98 |43.46$\pm$0.87 |50.52$\pm$1.95 |57.77$\pm$3.30 |
> | Mihi-2 | 58.75$\pm$0.39 |48.03$\pm$1.21 |47.89$\pm$1.37 |38.16$\pm$1.22 |39.16$\pm$1.60 |49.65$\pm$2.01 |
>
> _Table 2: Dynamics decoding accuracy across different random initializations._
>
> ### **Other action items**
>
> 1. **“The difference between (i) removing the alignment term but including the BlockSwap, and (ii) removing the alignment term and the original reconstruction term, is not clear”**
>
> * Referring to equations (2) and (3), our loss has four components: The usual KL-regularization in the style space and reconstruction loss used in VAEs, the swapped reconstruction loss, and a content space alignment term. In (i) we omit the content space alignment term, and in (ii) we omit both the usual reconstruction loss and the content space alignment term. We will be happy to revise this passage for clarity.
>
> 2. **“Clarity of the text of the multi-task disentanglement score”**
>
> * Thank you for your feedback on the clarity of the text. Upon re-reading we see how the definition of multiple tasks (yc and ys, lines 163-164) and disentanglement metrics can be improved.
>
> 3. **“Fig. 3D: bars should start at zero, not at 0.3”**
>
> * Thank you for your feedback on the figure. We will edit Fig. 3D to make the bars start at zero instead of 0.3.
>
> 4. **“Fig. 4D: y-axis is labeled “VAE”. Is this beta-VAE or vanilla VAE?”**
>
> * We are benchmarking the model’s generating performance with VAE instead of beta-VAE, following pi-VAE’s benchmarking method [2]. Because increasing beta introduces regularization that tends to reduce the overall reconstruction accuracy [3], we decided that it made sense to use a vanilla VAE for this experiment. In the final paper, we will also include results for the beta-VAE in the Appendix.
>
> **Refs**
>
> [1] Azabou, M., et al., “Mine your own view: Self-supervised learning through across-sample prediction.” 2021.
>
> [2] Zhou, D. and Wei, X.X., “Learning identifiable and interpretable latent models of high-dimensional neural activity using pi-VAE.” 2020.
>
> [3] Higgins, I., et al., “beta-vae: Learning basic visual concepts with a constrained variational framework.” 2016.

---

> > ### Author Response · Authors · 2021-08-10
> > **Additional response to Reviewer r6nv**
> >
> > ### **Additional comments**
> >
> > 1. **“The meaning of content vs. style could be discussed a bit better.”**
> >
> > * We use the terms “content” and “style” in two different contexts. The conceptual definition is that “content” of an example is the information pertaining to some decoding (e.g. the reach direction) and “style” is all the remaining information needed to reconstruct the observed neuronal activity. In our implementation, the “content” variables are those regularized by the alignment term, while the “style” variables are those regularized by the Gaussian prior. Thus, there is a mismatch in these definitions, because the conceptual “content” is with respect to the semantics of what we want to decode, while the “content” variables are trained without this knowledge. Our swapping scheme based on pairs of temporally-close examples is bound to incorporate some temporal variability.  (We did not intend to avoid this fact, which can be seen in Figure 3D without referring to the supplementary material.) While our model provides better disentanglement than other models (both generative models and discriminative models), we do agree that disentanglement without using labels remains a challenging task, and our method provides an initial step toward it. We will make sure we discuss this more openly in the discussion section.
> >
> > 2. **“Why treat decoding as a classification problem instead of regression?”**
> >
> > * While it is true that with the behavioral readouts (of the actual limb trajectory) are continuous, in this specific experiment, the actual cue that is given is a specific discrete target and thus we believe that classification is an informative measure of whether the target can be decoded from a learned latent space. In an ideal case, the angular variability in reach will have been effectively removed from the embeddings if the information is perfectly disentangled.
> >
> > 3. **“it is not clear to me what aspect of the loss function/training procedure would encourage such behavior”**
> >
> > * Our alignment term encourages positive examples (nearby points in time) to be mapped to similar points in the latent space. Thus, there is nothing in this loss that encourages circular structure. On the other hand, we regularize the style space with a Gaussian prior, which will impose some structural constraints on the decomposition. Experimentally, we find that in cases where we know the underlying latent structure (center-out reaching task, and synthetic experiment), Swap-VAE does a good job of preserving the global structure of the latent space. We will also revise the writing to make sure that we do not overemphasize this point and show new examples to support our claims.
> >
> > 4. **“In ‘We obtain a good denoised estimate of neural activity that is more indicative of what aspects of neural responses are stable and related to the movement tasks (rather than noise).’ I am not sure if noise is the right term here.”**
> >
> > * This is a fair point, we will edit to clarify that by “noise” we mean “factors extraneous to the particular classification task”.
> >
> > 5. **“Why not just use the SwapVAE latent rather than training a supervised classifier from generated samples?”**
> >
> > * We would like to show the possible usage of our model through this paragraph. Since our method provides a generative model that could separate the content space and the style space, it provides a possible approach to generate good data augmentations, which can be used to improve the performance of other downstream tasks.
> >
> > 6. **“The claim that the disentanglement for Chewie neural activity is on average lower than the Mihi neural activity seems like pure speculation.”**
> >
> > * Thank you for your suggestion. This claim is based on the multi-task covariance scores that are computed and plotted in Fig 3E. We are aware that this can be misunderstood, and will edit it accordingly to ensure our claims are not overstated in the revision.
> >
> > **Refs**
> >
> > [1] Azabou, M., et al., “Mine your own view: Self-supervised learning through across-sample prediction.” 2021.
> >
> > [2] Zhou, D. and Wei, X.X., “Learning identifiable and interpretable latent models of high-dimensional neural activity using pi-VAE.” 2020.
> >
> > [3] Higgins, I., et al., “beta-vae: Learning basic visual concepts with a constrained variational framework.” 2016.

---

> > > ### Comment · Reviewer_r6nv · 2021-09-01
> > > **Thorough response**
> > >
> > > Thank you for the thorough response and additional experiments. This addresses my main concerns. I am confident the promised revisions will improve the paper and changed my score accordingly.

---

### Official Review · Reviewer_hEnc · 2021-07-16

**Rating:** 7
**Confidence:** 3

**Summary:**

This work proposes a method for learning disentangled latent variables to generate neural activity. The proposal consists of (1) a self-supervised VAE that separates the latent spaces into vectors for for 'content', which is preserved across transformations, and 'style', which is altered across transformations; (2) a training augmentation which encourages content vectors to be interchangeable (swappable). The authors then propose a metric to measure the disentanglement between two variables. They illustrate properties of their proposed method using both synthetic experiments that generate data from ground truth latent variables and real neural data from a monkey reaching task, and compare them to other related methods in the literature. For the monkey reaching task, they find that disentanglement between the two latent spaces is higher for the monkey that needs to insert a delay in its response, suggesting that this method could provide insight into how latent variables are encoded under different task conditions.

**Limitations And Societal Impact:**

My questions about possible limitations are included in the main review above.

**Main Review:**

The description of the method and results are quite clear. However, I have questions about how well the proposed method compares to prior work. I will note that I am not very familiar with the related methods and do not have a good judgment on the significance of this contribution compared to that work. But I can see how a method for separating neural representations into 'content' and temporal dynamics (style) is clearly useful for neuroscience. On those terms I think the contribution is well worth accepting this paper. I also have some questions about the latent 'style' vector for the neural data.

- The decoding results (Fig 3D) seem to indicate that the supervised model does quite well at capturing elapsed time, performing better than the latent style space. From my understanding this is the case even though the supervised model is only trained to predict the reach locations. Can the authors comment on why this occurs?
    - This seems related to the fact that the content space also does reasonably well at decoding time. Both pieces of evidence suggest that disentanglement with the proposed approach is incomplete. This seems like a limitation of the current work -- do the authors have a proposal for how to address this?
- Table 1 shows accuracy in classifying reach direction for the proposed method and related approaches. It is not clear whether the differences are significant, especially because there are no reported error bars or statistical tests. (I do not see them in the Appendix either.) This could be easily done by looking at the accuracy over different validation folds, or over random seeds.
- Can the authors comment on how the 'style' latent space which captures temporal dynamics might be affected by temporal noise in neural activity? This method seems like it could be useful to create generative models of data where the sources of temporal variability are less well-understood, but the implications here are not clear to me.

Minor comments
- The monkeys are introduced by initials in the text, but referred to by their full names later. I would recommend standardizing this to avoid confusion when reading the figures (e.g. either write Monkey C everywhere, or Chewie everywhere).
- Fig 2: please include descriptions of the solid black and dotted lines in the legend.
- Table 1: would be useful to highlight/bold the best model(s) for each dataset
- Fig 4A/B: there are no x-axis tick labels
- Fig 4C/D: axes are not well-labeled to distinguish the two plots from each other. Could write "reach RMSE" and "time RMSE" on the plots

**Time Spent Reviewing:**

2

---

> ### Author Response · Authors · 2021-08-10
> **Response to Reviewer hEnc**
>
> Thank you for your positive review and excellent questions. We provide clarifications and describe new results that we have generated to address your questions.
>
> 1. **“The decoding results (Fig 3D) seem to indicate that the supervised model does quite well at capturing elapsed time, performing better than the latent style space.”**
>
> * We would like to clarify that in our evaluations, we train two different supervised models, one trained to decode reach direction and the other trained to decode time. The results in Fig. 3D are provided for each of their corresponding supervised models, and that is the reason why the supervised model performs well at capturing elapsed time. We will make sure that this is clear in the revision.
>
> 2. **“The content space also does reasonably well at decoding time … suggest that disentanglement with the proposed approach is incomplete… do the authors have a proposal for how to address this?”**
>
> * It is a challenging problem to develop a useful notion of content without using labels. Here, we implicitly assume that a pair of examples share the same content if they are temporally close together and thus we often have some information about time in the content space and vice versa. One possible solution is to increase the weight of the alignment term in the loss function. While developing the method, we tried to increase the alignment loss to encourage the disentanglement further, but found that doing this would impact the reach decoding performance as well and thus fixed the proportion of the two terms in favor of simplicity. We expect that by tuning the contributions of the alignment loss with the generative terms would result in better separation between the two components. We will include a discussion along these lines in the revised text.
>
> 3. **“Table 1 - It is not clear whether the differences are significant, especially because there are no reported error bars or statistical tests.”**
>
> * Please see our general response where we ran additional experiments to obtain the mean and standard deviation over different random initializations of our model and competitors. Our results provide evidence that Swap-VAE is stable and outperforms other approaches in terms of their decoding accuracy.
>
> 4. **“Can the authors comment on how the 'style' latent space which captures temporal dynamics might be affected by temporal noise in neural activity?”**
>
> * Thank you for your great question. While different sources of variability (or noise) can be contained in both the content and style space, our content space is actively trying to build temporal invariance by comparing points that are close in time and encouraging them to essentially have the same representation (through swapping). The style space, on the other hand, contains the remaining variability (Gaussian prior) needed to reconstruct the neural activity. Thus, we expect that temporal variability that occurs over short timescales will be “removed” in the content space and will instead be incorporated into our style or generative space.
>
> 5. **Minor comments**
>
> * Thank you for catching this inconsistency in how we refer to the individuals that we have tested our approach on. We have made all usage consistent. We will also make changes to the figures and tables as you’ve suggested.

---

### Official Review · Reviewer_sJZd · 2021-07-25

**Rating:** 7
**Confidence:** 4

**Summary:**

The study proposed an unsupervised (self-supervised) learning strategy to model neural activity in terms of meaningful latent factors. The authors apply the proposed method to both synthetic and actual neural datasets and show that their approach can both successfully disentangle neural activity along meaningful behaviorally-associated latent factors, and also generate realistic patterns of neural activity.  A significant methodological advance is that the disentanglement is achieved in the learned latent space in an entirely unsupervised manner, without requiring any attribute labeling. While some evaluation decisions could be criticized, the results (and success of the approach) are overall convincing.

**Limitations And Societal Impact:**

The paper is overall interesting, well-written and makes a valuable contribution. I do, however, have some comments for the authors to consider (which in my mind, are potential limitations of the study):
-	Comparison of the proposed unsupervised method with the supervised baseline is not suggestive because of the absence of augmentations in the supervised baseline. The authors should consider reporting performance on the decoding task when the supervised method employed data augmentations as well.
-	For completeness, the authors should also report how the hyperparameters for the linear decoder were determined. Ideally, I would’ve liked to see error bars for decoding accuracies as well (maybe by bootstrapping training set for the decoder?)
-	In future, the authors could also consider replacing the acc metric for decoding with better evaluation metrics for circular data, like circular correlation. This would treat the reach direction as a continuous variable (which it is) rather than as a discrete unordered variable (which it theoretically isn’t).
-	The authors consider swapping only the block of variables belonging to the `content’ group. What would happen if the reconstruction term in the BlockSwap method swapped both the content and style of the augmented views? Does swapping only the content block necessarily facilitate disentanglement? If the BlockSwap was essential, does the proposed method required knowing the number of latent factors in advance. The authors could discuss these aspects in their conclusion/discussion.
-	When comparing the proposed SwapVAE against vanilla VAE, the authors should also consider reporting other metrics more commonly employed in VAE evaluation (likelihood etc.) and not just the reconstruction error (which can be trivially minimized). This is important, since the authors mention ‘generating realistic neural activity’ as a significant contribution of their paper.
-	The authors should also consider defining content and style more broadly as it relates to their specific neural application (e.g., as in Gabbay &Hosehn (2018)) where style is instance-specific(?) and content includes information that can be transferred among groups. More specifically, since their model is not sequential and does not capture the temporal dynamic structure, what do they really mean by ‘style’ represents the ‘movement dynamic’?


**Main Review:**

This is an interesting paper where the methods (associated loss functions) are clearly stated and explained, and the described analyses seems well-executed. I have some comments about the lack of thorough analysis, which potentially prevent drawing strong conclusions about the benefits of different model components in driving the learned representations. Although, that said, I think the paper does pack a lot in 8 pages, so the lack therein could be attributed to too much content. I am not sure that the proposed approach extends previous methods for generative modeling (and disentanglement) significantly. Although coupled with the specific application (modeling ‘of neural activity’), the combination of methods employed (contrastive learning + generative modeling) in this study could be considered novel. The application of the proposed method for neural datasets is well-motivated; specifically, the augmentations and other decisions seem like good choices for neural representations. Visualization and other quantitative analyses on the synthetic datasets clearly illustrate the merits of the proposed approach and the results on the real neural datasets are promising for future applications geared towards discovering the structure in dynamic patterns of neural activity.

**Time Spent Reviewing:**

4-5 hours

---

> ### Author Response · Authors · 2021-08-10
> **Response to Reviewer sJZd**
>
> Thank you for your positive review and excellent questions. We provide clarifications and new results that we have generated to address your questions below.
>
> 1. **Clarification about the supervised baseline**
> * We would like to clarify that our supervised baseline used data augmentation as well. We tested the supervised model both with and without data augmentation (DA), and due to space constraints, we included only the better (with DA) of the two. We will make this point clearer in the revision.
>
> 2. **Hyperparameters for the linear decoder and error bars for decoding accuracies**
>
> * Thank you for your suggestions. In Table 1 below, we report the mean and standard error obtained by our method and compare with the reach decoding accuracies reported in [1] for MYOW and BYOL. We also computed the mean and SE for the pi-VAE, the beta-VAE, and BYOL for comparison. We report the accuracies for our model in the last two columns, where Ours (1) denotes the result we obtain when we train and evaluate networks with 5 different random seeds, and in Ours (2) we further select the model with the best performance and then retrain the linear decoding layer training from 5 random initializations. Our results confirm that Swap-VAE maintains a gap over other methods and provides insights into the different sources of variance in our model. These results will be provided in the revised manuscript.
>
> * For the hyperparameter selection of the linear decoder, we first did a grid search of learning rate and weight decay based on the supervised pre-trained feature space, and used the same set of learning rate and weight decay throughout. We will include this information in the Appendix to make it clearer.
>
> |         |   | Supervised | pi-VAE | $\beta$-VAE | BYOL (repro) | BYOL [1] | MYOW [1] | Ours (1) | Ours (2) |
> |---------|-------|-------| -------| -------| -------| -------| -------| -------|  -------|
> | Chewie-1 | acc | 61.59$\pm$2.04 | 66.30$\pm$1.30 |64.34$\pm$1.08 |62.12$\pm$2.27 | 63.80 | 70.41 | 72.81$\pm$1.40 | 74.47$\pm$0.27 |
> |  | $\delta$-acc | 77.58$\pm$1.28 | 82.93$\pm$1.34 |80.83$\pm$1.08 |81.27$\pm$1.08 |81.90 |86.24  |85.04$\pm$0.94 |86.25$\pm$0.26 |
> | Chewie-2 | acc | 69.71$\pm$1.85 | 61.33$\pm$0.86 |60.24$\pm$3.18 |57.25$\pm$1.11 | 57.17| 60.95 |68.97$\pm$3.41 |75.37$\pm$0.62 |
> |  | $\delta$-acc | 78.18$\pm$2.96 |73.63$\pm$1.80 |80.09$\pm$1.75 |76.89$\pm$1.30 | 77.36| 81.36 |83.66$\pm$1.81 |86.12$\pm$0.29 |
> | Mihi-1 | acc | 62.86$\pm$1.08 |62.63$\pm$1.42 |58.11$\pm$1.52 |60.03$\pm$0.95 | 59.50 | 70.48 |64.26$\pm$0.89 | 66.00$\pm$0.71 |
> |  | $\delta$-acc | 79.10$\pm$1.20 |79.20$\pm$2.30 |75.98$\pm$1.10 |78.82$\pm$1.68 |79.78 |83.24 |82.18$\pm$1.55 |83.31$\pm$0.66 |
> | Mihi-2 | acc | 60.72 $\pm$1.74 |62.70$\pm$0.90 |60.23$\pm$0.96 |59.94$\pm$1.39 |60.82 |64.35 |66.12$\pm$0.87 |66.94$\pm$0.58 |
> |  | $\delta$-acc | 74.02$\pm$3.18 |76.89$\pm$1.28 | 77.89$\pm$1.18 |78.10$\pm$1.85 | 78.30 | 80.58 | 82.74$\pm$0.77 | 84.24$\pm$0.11 |
>
> _Table 1: Reach decoding accuracy across different random initializations_
>
> 3. **“In future, the authors could also consider replacing the acc metric for decoding with better evaluation metrics for circular data, like circular correlation. This would treat the reach direction as a continuous variable (which it is) rather than as a discrete unordered variable (which it theoretically isn’t).”**
>
> * While it is true that the behavioral readouts (of the actual limb trajectory) are continuous, in this specific experiment, the actual cue that is given is a specific discrete target and thus we believe that this is an informative measure of whether the target can be decoded from a learned latent space. In an ideal case, the angular variability in reach will have been effectively removed from the embeddings if the information is perfectly disentangled.
>
> 4. **“What would happen if the reconstruction term in the BlockSwap method swapped both the content and style of the augmented views?”**
>
> * This amounts to reconstructing an example given the latent representation of a near neighbor, which would eliminate the variation in the style variables we are trying to preserve. We tested it when doing ablation experiments and found that VAE with the alignment term added to its loss gives poor performance.
>
> 5. **“Does swapping only the content block necessarily facilitate disentanglement?”**
>
> * That is a great question. To answer it, we provide the disentanglement score for all of the ablations that we tested (see below table 2). Our results suggest that keeping the alignment loss and the original reconstruction term but removing the BlockSwap augmentation (No-Swap) does not facilitate better disentanglement compared with beta-VAE. Removing the alignment term but including the BlockSwap (no L2) provides better disentanglement. Removing the alignment term and the original reconstruction term (Swap-only) provides the closest disentanglement score as our model. Swap-VAE, which contains a combination of alignment term, BlockSwap, and the original reconstruction term, provides the best disentanglement performance. Thus, we believe our approach is well-suited to disentangling semantic information while preserving individual variance.
>
> |        | Ours | $\beta$-VAE | no L2 | No-Swap | Swap-only |
> |--------|-------| -------| -------| -------| ------- |
> | Multi-task disentanglement score | 0.5380 |0.2070 |0.3455 |0.2089 |0.4914 |
>
> _Table 2: Multi-task disentanglement score over different ablations._
>
> 6. **“If the BlockSwap was essential, does the proposed method require knowing the number of latent factors in advance.”**
>
> * That is a great question and we will include this in discussion. The proposed method does not necessarily require knowing the number of latent factors, but different separation of the content and style subspaces does give different model performance. We included our ablation across different dimensions of the content and the style space inside the appendix section D.1. Our model performs reasonably well across different dimensions. Also, this additional hyperparameter space can be searched based on a train/validate/test split during the training.
>
> 7. **When comparing the proposed SwapVAE against vanilla VAE, the authors should also consider reporting other metrics more commonly employed in VAE evaluation (likelihood etc.)**
>
> * Thank you for the suggestion. We are still investigating better metrics to evaluate our methods.
>
> 8. **“The authors should also consider defining content and style more broadly as it relates to their specific neural application (e.g., as in Gabbay &Hosehn (2018)) where style is instance-specific(?) and content includes information that can be transferred among groups.”**
>
> * Thank you for the great suggestion. We will include additional examples of how this same idea may be applied in other contexts in the discussion.
>
> 9. **“More specifically, since their model is not sequential and does not capture the temporal dynamic structure, what do they really mean by ‘style’ represents the ‘movement dynamic’?”**
>
> * We only use “content” and “style” as an analogy to the concepts in computer visione define content as what is consistent across subsets of neurons and samples in time (for example, during a particular reach) and the style is the remaining variability needed to reconstruct the original neural activity (that is, to generate data). We find that in our exemplary reaching task, this corresponds to decomposing the reach direction and the dynamics.
>
> **Refs**
>
> [1] Azabou, M., et al., “Mine your own view: Self-supervised learning through across-sample prediction.” 2021.

---

### Author Response · Authors · 2021-08-10
**General Response**

We would like to thank all reviewers for their valuable feedback and suggestions. We look forward to incorporating these suggestions in a revised version of the paper. We also very much appreciate the positive feedback which described our work as a significant methodological advance (sJZd), convincing (sJZd), novel and promising (sJZd, r6nv), clearly useful (hEnc), well-motivated (sJZd), and clear (sJZd, hEnc). Responses to reviewer feedback are provided for each review independently.

### **New experiments to test the stability of our approach**
To address reviewer concerns over the robustness and stability of our results, we have run a number of new experiments to study our model performance over different random initializations (see Table A below). We computed the accuracy of reach decoding for a supervised model, pi-VAE, beta-VAE, BYOL, and our model for 5 random different random seeds, and included the mean and standard deviation inside the table. We report the accuracies for our model in the last two columns, where *Ours (1)* denotes the result we obtain when we train and evaluate networks with 5 different random seeds, and in *Ours (2)* we further select the model with the best performance and then retrain the linear decoding layer training using 5 random initializations. Our results confirm that Swap-VAE maintains a gap over other methods and provides insights into the different sources of variance in our model. These results will be provided in the revised manuscript.

|         |   | Supervised | pi-VAE | $\beta$-VAE | BYOL (repro) | BYOL [1] | MYOW [1] | Ours (1) | Ours (2) |
|---------|-------|-------| -------| -------| -------| -------| -------| -------|  -------|
| Chewie-1 | acc | 61.59$\pm$2.04 | 66.30$\pm$1.30 |64.34$\pm$1.08 |62.12$\pm$2.27 | 63.80 | 70.41 | 72.81$\pm$1.40 | 74.47$\pm$0.27 |
|  | $\delta$-acc | 77.58$\pm$1.28 | 82.93$\pm$1.34 |80.83$\pm$1.08 |81.27$\pm$1.08 |81.90 |86.24  |85.04$\pm$0.94 |86.25$\pm$0.26 |
| Chewie-2 | acc | 69.71$\pm$1.85 | 61.33$\pm$0.86 |60.24$\pm$3.18 |57.25$\pm$1.11 | 57.17| 60.95 |68.97$\pm$3.41 |75.37$\pm$0.62 |
|  | $\delta$-acc | 78.18$\pm$2.96 |73.63$\pm$1.80 |80.09$\pm$1.75 |76.89$\pm$1.30 | 77.36| 81.36 |83.66$\pm$1.81 |86.12$\pm$0.29 |
| Mihi-1 | acc | 62.86$\pm$1.08 |62.63$\pm$1.42 |58.11$\pm$1.52 |60.03$\pm$0.95 | 59.50 | 70.48 |64.26$\pm$0.89 | 66.00$\pm$0.71 |
|  | $\delta$-acc | 79.10$\pm$1.20 |79.20$\pm$2.30 |75.98$\pm$1.10 |78.82$\pm$1.68 |79.78 |83.24 |82.18$\pm$1.55 |83.31$\pm$0.66 |
| Mihi-2 | acc | 60.72 $\pm$1.74 |62.70$\pm$0.90 |60.23$\pm$0.96 |59.94$\pm$1.39 |60.82 |64.35 |66.12$\pm$0.87 |66.94$\pm$0.58 |
|  | $\delta$-acc | 74.02$\pm$3.18 |76.89$\pm$1.28 | 77.89$\pm$1.18 |78.10$\pm$1.85 | 78.30 | 80.58 | 82.74$\pm$0.77 | 84.24$\pm$0.11 |

_Table A: Reach decoding accuracy across different random initializations_


**Refs**

[1] Azabou, M., et al., “Mine your own view: Self-supervised learning through across-sample prediction.” 2021.

---

### Decision · Program_Chairs · 2021-09-27

**Decision:**

Accept (Oral)

**Comment:**

This paper present a method for learning disentangled latent variables inherent in brain activity, and illustrate their methods with synthetic and real monkey data from a reaching task. After the initial response from the reviewers, the authors responded to their concerns extensively, including running additional experiments and addressing the concerns of the reviewers. Given that all the reviewers expressed that their concerns were addressed by the responses, I recommend acceptance and ask the authors to modify the paper to include these changes and address all the comments from the reviewers.